# Cellulose Paper Modified by a Zinc Oxide Nanosheet Using a ZnCl_2_-Urea Eutectic Solvent for Novel Applications

**DOI:** 10.3390/nano11051111

**Published:** 2021-04-25

**Authors:** Changmei Lin, Duo Chen, Zifeng Hua, Jun Wang, Shilin Cao, Xiaojuan Ma

**Affiliations:** 1College of Materials Engineering, Fujian Agriculture and Forestry University, Fuzhou 350002, China; 15980205608@163.com (C.L.); 13459735703@163.com (Z.H.); wangjun20201118@163.com (J.W.); 2International School for Optoelectronic Engineering, Qilu University of Technology (Shandong Academy of Science), Jinan 250353, China; sdkdcd@163.com

**Keywords:** cellulose paper, zinc oxide, triboelectric nanogenerator, antibacterial activity, pressure sensor

## Abstract

Cellulose paper has been functionalized by nanoparticles such as Ag nanoparticles, TiO_2_, and BaTiO_3_ for versatile applications including supercapacitor, sensors, photoactivity, and packaging. Herein, zinc oxide (ZnO) nanosheet-modified paper (ZnO@paper) with excellent antibacterial properties was fabricated via a mild ZnCl_2_-urea eutectic solvent. In this proposed method, cellulose fibers as the raw material for ZnO@paper were treated by an aqueous solvent of ZnCl_2_-urea; the crystalline region was destroyed and [ZnCl]^+^-based cations were adsorbed on the surface of cellulose fibers, facilitating more ZnO growth on ZnO@paper. A flexible paper-based triboelectric nanogenerator (P-TENG) was made of ZnO@paper paired with a PTFE film. The P-TENG presents high triboelectric output performance and antibacterial activity. For instance, the output voltage and current of the P-TENG were 77 V and 0.17 μA, respectively. ZnO@paper showed excellent antibacterial activity against *E. coli* and *S. aureus*, suggesting that a P-TENG can restrain and kill the bacteria during the working process. The results also indicated that ZnO could improve the surface roughness of cellulose paper, enhancing the output performance of a flexible P-TENG. In addition, the potential application of a P-TENG-based pressure sensor for determining human motion information was also reported. This study not only produced a high-performance P-TENG for fabricating green and sustainable electronics, but also provides an effective and novel method for ZnO@paper preparation.

## 1. Introduction

Cellulose paper, a common commodity in daily life, has been extensively used in the fields of food package, supercapacitors, lithium-ion batteries, and solar cells because it is abundant, biodegradable, lightweight, recyclable, and environmentally friendly [1,2,3,4,5,6,7]. In recent years, the applications of cellulose paper in nanogenerators such as triboelectric nanogenerators and piezoelectric nanogenerators have also been reported [8,9].

Triboelectric nanogenerator (TENG) is a novel technology for harvesting mechanical energy from human motions, sounds, and structure vibrations, and has attracted much attention due to its wide availability [10,11]. It is a device that utilizes electrostaticity and triboelectricity to convert mechanical energy into electricity, and its fundamental theoretical origin is Maxwell’s displacement current [12,13,14,15]. It also has been intensely followed with interest among researchers because TENG has the advantages of low cost, high energy conversion, abundant material choices, and a simple fabrication process.

As an electron-donating material, cellulose paper has already been employed to construct TENG, which has the advantages of being cheap, lightweight, flexible, eco-friendly, and disposable [8,16,17]. However, traditional papers used to prepare TENG, such as hard paper, print paper, paper cards, rice paper, and crepe paper, are commercial and non-antibacterial, limiting the application of a P-TENG due to the low flexibility and incapability against bacteria of these papers [8,18,19,20,21,22]. Besides, in the fabrication process of P-TENG, the surface modification of cellulose fibers to prepare cellulose paper is always neglected. Therefore, it is vital to prepare a P-TENG using composite paper that is composed of surface-modified cellulose fibers.

On the other hand, ZnO is a semiconductor that possesses a wide bandgap (3.37 eV) and high exciton binding energy (60 meV) [23]. It has been widely used in the field of piezoelectric devices, optical devices, solar cells, transparent electrodes, sensors, photocatalysts, and antiseptics due to the unique characteristics of ZnO [24,25,26,27,28,29,30]. To prepare a ZnO-modified cellulose composite, one method is to disperse ZnO nanoparticles into a cellulose suspension [31,32,33]. However, due to the low bonding strength between ZnO and cellulose fibers, ZnO would detach from the cellulose fibers, causing environmental pollution during the application of a ZnO-modified cellulose composite. Another method is to immerse the paper or cellulose fibers of which the paper is made into the solutions containing Zn^2+^ (ZnO precursor) such as zinc chloride (ZnCl_2_), zinc nitrate (Zn(NO_3_)_2_), zinc acetate (Zn(CH_3_COO)_2_), or zinc sulfate (ZnSO_4_), and the paper is treated with sodium hydroxide (NaOH), potassium hydroxide (KOH), or ammonium hydroxide (NH_4_OH), inducing the nucleation and growth of a ZnO crystal on cellulose fibers [34,35,36,37,38]. In spite of the successful preparation of a ZnO-modified cellulose composite by the latter method, the fabrication process was complicated, and the ZnO deposited on paper or cellulose fibers was low due to the poor accessibility between the ZnO precursor and cellulose fibers (namely, the ZnO precursor is difficult to get to the hydroxyl groups on cellulose), limiting the ZnO-modified cellulose paper to industrial production. In consequence, it is significant to develop a new route for preparing ZnO-functionalized cellulose paper. Furthermore, as mentioned, ZnO@paper has versatile applications, for example, as sensors and packaging [7,39], but there are few reports concerning ZnO@paper used as a friction layer for a P-TENG.

In this work, we report a novel method for the preparation of ZnO@paper. The preparation mechanism of ZnO@paper is investigated. Furthermore, the potential application of ZnO@paper in a P-TENG for energy harvesting and as a pressure sensor is also demonstrated. 

## 2. Materials and Methods

### 2.1. Materials

Dissolving pulp with cellulose content of 91.9 wt.% and hemicellulose content of 8.1 wt.% prepared from softwood was provided by Qingshan Paper Co., Ltd. (Sanming, China). Zinc chloride (ZnCl_2_, 96%) and urea (CO(NH_2_)_2_, 99%) were purchased from Sinopharm Chemical Reagent Co., Ltd. (Shanghai, China) and Aladdin (Shanghai, China), respectively. Agar, *Escherichia coli*, and *Staphylococcus aureus* were provided by Qingdao Haibo Biological Technology Co., Ltd. (Qingdao, China). 

### 2.2. Preparation of ZnO@paper

#### 2.2.1. ZnO@paper Preparation Process

Dissolving pulp was dispersed using a homogenizing and defibration device (ZQS4, Xianyang Tongda Light Industrial Equipment Co., Ltd., Xianyang, China). The eutectic solvent composed of ZnCl_2_ and CO(NH_2_)_2_ was fabricated at 110 °C with an amount of substance ratio of 1:3 by magnetic stirring. After melting the ZnCl_2_ and CO(NH_2_)_2_, the eutectic solvent was cooled to 55 °C. Then 500 mL distilled water and 10 g pulp were added into the eutectic solvents and stirred 10 min to form suspension at 55 °C. Later, the temperature was increased to 85°C for the nucleation and growth of ZnO. After 2 h, the reaction product was filtered and washed with distilled water until the Cl^−^ was no longer present in the filter. Finally, ZnO@paper with a grammage of 60 g/m^2^ was fabricated using a papermaking device equipped with a filter device of 10 cm radius (FRANK-PTI, S958540014, Birkenau, Germany). The grammage of the ZnO@paper is expressed as the mass of ZnO-modified cellulose fibers divided by the filter area of the papermaking device. Then the ZnO@paper was dried by squeezing, using the drying device of the papermaking device under pressure from a heavy plate. Meanwhile, blank paper was also prepared from cellulose fibers without ZnO modification. To control the mass of ZnO deposited on the ZnO@paper and according to the mass ratio of ZnO by fully formed Zn^2+^ to the cellulose fibers, the amount of ZnCl_2_ was set at 0.01 mol, 0.03 mol, 0.05 mol, 0.07 mol, and 0.1mol, respectively. The amount of CO(NH_2_)_2_ was also adjusted with the amount of ZnCl_2_ to maintain the 1:3 ratio. The samples were marked as ZnO@paper-1, ZnO@paper-2, ZnO@paper-3, ZnO@paper-4, and ZnO@paper-5.

#### 2.2.2. The Mechanism of ZnO@paper Preparation

It has been reported that the high degree of crystallinity of cellulose is a barrier for sufficient ZnO assembling on cellulose fibers [34,37,40]. Thus, destroying the crystallinity region is beneficial for more ZnO growing on cellulose fibers. The ZnCl_2_-urea eutectic solvent consists of [ZnCl]^+^-based cations including [ZnCl (urea)]^+^, [ZnCl (urea)_2_]^+^, [ZnCl(urea)_3_]^+^, and anion [ZnCl_3_]^−^ [41,42]. When water is added into the ZnCl_2_-urea eutectic solvent, the urea composition is dissolved and the hydrogen bonds are destroyed; resulting in the disruption of the structure of the above cations and, therefore, ZnO precipitation. The chemical reaction is shown as follows:(1)$$\mathrm{CO}{\left({\mathrm{NH}}_{2}\right)}_{2}+{\;H}_{2}O\overset{\Delta }{\to }2{\mathrm{NH}}_{3}↑+{\;\mathrm{CO}}_{2}↑$$
NH_3_ + H_2_O → NH_4_^+^ + OH^−^(2)
[ZnCl]^+^ + 4OH^−^ → Zn(OH)_4_^2−^ + Cl^−^(3)
[ZnCl_3_]^−^ + 4OH^−^ → Zn(OH)_4_^2−^ + 3Cl^−^(4)
Zn(OH)_4_^2−^ → ZnO + H_2_O + OH^−^(5)

Inspired by the above reaction sequence, we designed a route for ZnO@paper preparation. In the first step, ZnCl_2_ and CO(NH_2_)_2_ were mixed at 110 °C until they were completely melted. Next, distilled water and cellulose fibers were added subsequently into the above ZnCl_2_-urea eutectic solvent; in this instance, the [ZnCl]^+^-based cations and anion [ZnCl_3_]^−^ reacted with the OH^−^ from urea dissolving in water by thermal treatment, and the [ZnCl]^+^ and [ZnCl_3_]^−^ transforming into Zn(OH)_4_^2−^. The Zn(OH)_4_^2−^ further converted into ZnO owing to the dehydration reaction. Hence, ZnO was deposited on the surface of cellulose fibers by hydrogen bonding with hydroxyl groups of cellulose. Finally, the paper was modified by ZnO with the aid of the hydrogen bonding between hydroxyl groups on cellulose fibers and ZnO, as illustrated in Figure 1. The [ZnCl]^+^-based cations would be adsorbed on their surface because of the negative charges of cellulose fibers. Besides, the crystalline region would be destroyed by the OH^−^. Thus, ZnCl_2_-urea could facilitate more ZnO growing on the surface of cellulose fibers.

### 2.3. Integration of the Arch-Shaped P-TENG 

To fabricate the P-TENG, at first, ZnO@paper (thickness of 125 μm) and polytetrafluoroethylene (PTFE) (thickness of 50 μm) with an area of 6 × 6 cm^2^ were coated with a thin gold film as the electrode by using sputtering (MC1000, Japan Hitachi Nake high-tech enterprise, Tokyo, Japan) in a vacuum and a current of 40 mA for 5 min, making their surface resistance below10 Ω. Secondly, the gold film sputtered onto the ZnO@paper was affixed at the center of print paper (A4 paper used for printing ) (7 cm × 8 cm) using 3M tape, while the PTFE film was also pasted onto print paper with an area of 7 cm × 10 cm adopting 3M tape. Finally, the edges of these two print papers were jointed together using Kapton tape for the reutilization of PTFE because Kapton tape was easy to be peel off, forming an arch-shaped device. The arch-shaped P-TENG could maintain a proper restoring force when it was subjected to an external force, which was beneficial for contact-separation of the P-TENG, as showing in Figure 2a.

The fabrication of the pressure sensor was similar to that of the P-TENG except that the effective size of the ZnO@paper and PTFE was 3 cm × 3 cm. The ZnO@paper and PTFE were pasted onto print paper with an area of 4 cm × 5 cm and 4 cm × 7 cm, respectively.

The operating principle of the P-TENG is shown in Figure 2b [43,44]. When the ZnO@paper and PTFE film are brought into contact with each other, electrons are injected from the ZnO@paper to the PTFE film, leading to the surface of the ZnO@paper obtaining positive charges while the PTFE film obtains negative charges, which are named as triboelectric charges. When the gap between ZnO@paper-4 and PTFE became zero, the triboelectric charges on the surfaces of the ZnO@paper and PTFE film are in electrical equilibrium (Stage Ⅱ). At the disconnecting stage, the surfaces of the ZnO@paper and PTFE film would automatically diverge, and the triboelectric charges would generate an electrical field between the surfaces of the ZnO@paper and PTFE film, creating an electric potential difference between Au electrodes (Stage Ⅲ). When the separating distance reaches the biggest value in a full contact-separation cycle, it will achieve a new electrical equilibrium (Stage Ⅳ). Subsequently, once the ZnO@paper is compressed toward the PTFE surface again, the induced charges in two Au electrodes become unbalanced (Stage Ⅴ). As a result, the electrons flow between Au electrodes through an external circuit, producing an alternating signal voltage and current as the periodic mechanical force, as illustrated in Figure 2c,d).

### 2.4. Structure Characterization of ZnO@paper

X-ray diffraction patterns (XRD) was performed to show that the product formed on the ZnO@paper was ZnO rather than other zinc compounds by using an X-ray diffractometer (Ultima IV, Japan) with Cu Kα radiation (λ = 0.1542 nm) at 40 kV and 40 mA in the diffraction angle (2θ) range of 5–80°. The morphologies of the samples were recorded on a scanning electron microscope (SEM) (Verios G4 UC, FEI, Hillsboro, OR, USA) with elemental analysis observed by energy-dispersive X-ray spectroscopy (EDS) operating at 2 kV, and the samples were coated with a thin layer of gold using sputtering (MC1000, Japan Hitachi Nake high-tech enterprise, Tokyo, Japan) prior to the observation. The chemical bonding states of both cellulose fibers and the ZnO-forming ZnO@paper were determined by an X-ray photoelectron spectrometer (XPS) (ESCALAB 250; Thermo Scientific, Waltham, MA, USA).

### 2.5. Antibacterial Determination of ZnO@paper

In our study, *Escherichia coli* (typical gram-negative bacteria) and *Staphylococcus aureus* (a type of gram-positive bacteria) were used to assess the antibacterial properties of ZnO@paper. The culture medium was prepared by agar. In brief, 33 g of agar was dissolved in 1000 mL distilled water, and the solution was heated until it became transparent. Before the experiment, LB broth (lysogeny broth), the medium, and apparatus (including beaker, pipette tips, test tube, test tube holder, and so on) were sterilized using an autoclave (SYQ-DSX-2808, Shanghai Shenan Medical Instrument Factory, Shanghai, China) under a pressure of 0.1 MPa and the temperature was maintained at 121 °C for 15 min. Then *E. coli* and *S. aureus* with concentrations of 10^5^ CFU were added into 10 mL LB broth, respectively. Next, a 0.1 g sample cut into small pieces was added to the LB broth. Finally, the LB broth was cultivated for 24 h at 37 °C with shaking at 140 rpm. After cultivation, the bacterial suspensions were diluted by 10 to 10^7^ times by using 0.5 wt.% NaCl, and then 0.1 mL different dilutions were spread on the agar plates, and the dishes were placed in an incubator for 24 h at 37 °C. The antibacterial activity was evaluated by counting the number of colony-forming units (CFU) from the culture medium distributed uniformly by the bacterial suspensions diluted 100 times using the plate colony-counting method. To further show intuitively the antibacterial properties of ZnO@paper, the culture medium was also recorded by a camera. The antibacterial ratio was expressed as follows:(6)$$\mathrm{Antibacterial}\;\mathrm{ratio}\left(\%\right)=\frac{{\mathrm{CFU}}_{\mathrm{blank}\;\mathrm{paper}}\;{-\;\mathrm{CFU}}_{\mathrm{ZnO}@\mathrm{paper}}}{{\mathrm{CFU}}_{\mathrm{blank}\;\mathrm{paper}}}\;\times \;100\%$$

### 2.6. Measurements of the P-TENG as a Pressure Sensor

The open-circuit voltage (V_OC_) of the P-TENG was measured by an electrometer with an input resistance of 200 TΩ (6517B, Keithley). The short-circuit current (I_SC_) was observed using a low noise current preamplifier (SR570, Stanford Research Systems). A mechanical linear motor (HF01-37, Linmot) was employed to provide different external forces for driving the P-TENG. All measurements for the P-TENG were observed using a contact-separation frequency of 1 Hz. The response signal of the pressure sensor was carried out by an LCR meter (TH2832, Changzhou, China).

## 3. Results and Discussion 

### 3.1. Characterizations of ZnO@paper

To prove the successful formation of ZnO nanosheets in ZnO@paper, ZnO@paper-4 was used as the typical example for analysis.

The XRD spectra of blank paper and ZnO@paper-4 are given in Figure 3. As shown in Figure 3a, the spectrum of ZnO@paper-4 presents peaks at 15.6°, 22.6°, and 34.5°, assigning to (101), (002), and (004) crystallographic plane, respectively; it has a similar diffraction pattern to blank paper which is cellulose I [45]. Additionally, the crystalline index (CrI) of the ZnO@paper-4 is lower than blank paper becauseZnCl_2_ with a concentration lower than 65 wt.% can penetrate into the crystalline region of cellulose, swelling cellulose and decreasing the crystallinity of cellulose due to the high polarity [46,47]. Cellulose has many hydroxyl groups which can promote the formation of ZnO. However, the high CrI is a barrier for Zn^2+^ (ZnO precursor) to penetrate into crystallinity regions of cellulose, so that hydroxyl groups on cellulose cannot be fully utilized leading to it is very difficult to assemble much more ZnO on cellulose fibers [34,37,47]. Therefore, decreasing the CrI could facilitate more ZnO deposited on ZnO@paper. 

Furthermore, compared with blank paper, the XRD spectra of ZnO@paper-4 shows new characteristic peaks at 2θ value of 31.8° (100), 33.5° (002), 38.1° (101), 45.7° (102), 54.5° (110), 61.5° (103), 66.5° (200), 67.9° (112) and 69.7° (201), as illustrated in Figure 3b. The results suggest that ZnO in paper has a hexagonal structure [35,40]. However, the peaks of the ZnO@paper at 30–80° of 2θ value are weak. Therefore, further demonstration was completed using EDS and XPS.

It is apparent that ZnO@paper-4 is composed of C, O, and Zn elements while blank paper just contains C and O elements (Figure 4). The EDS results further suggests the successful preparation of ZnO@paper.

As a comparison, Table 1 lists the content of the Zn element in the ZnO/cellulose composite prepared by reported methods and our method. Zn content is in a range of 2.1–7.02% by using a precipitation and blending method. In our work, the Zn content is much higher than that of the other methods, which is up to 13.67%. The results confirmed that more ZnO assembled on cellulose fibers since the hydroxyl groups in cellulose could be fully utilized owing to the decrease of cellulose CrI [34,37,40].

The XPS analysis of blank paper and ZnO@paper-4 demonstrates the chemical bonding states to confirm the presence of ZnO in ZnO@paper. The survey spectra (Figure 5a) of blank paper and ZnO@paper-4 show that there is an adventitious Zn element in ZnO@paper-4, which is consistent with EDS results. The Zn2p regions of the XPS spectra (Figure 5b) indicate that ZnO@paper-4 has two peaks at 1020.7 eV and 1043.8 eV, which are attributed to Zn2p_3/2_ and Zn2p_1/2_ respectively, suggesting that Zn^2+^ exists in ZnO@paper-4 [49,50]. Furthermore, the binding energy gap between Zn2p_3/2_ and Zn2p_1/2_ is 23.15 eV, showing that the Zn atoms are in a completely oxidized state [51]. In addition, the O1s spectra of blank paper (Figure 5c) only consist of a peak centered at 531.3 eV, attributed to the C-O of cellulose, while the O1s spectra of ZnO@paper-4 deconvolution leads to two new signals at a binding energy of 529.4 eV corresponding to lattice oxygen and 532.5 eV attributed to vacancy oxygen [52,53,54]. For the C1s, the two samples can be deconvoluted into three peaks at ~284.5 eV, ~286.3 eV, and 287.5 eV, which are ascribed to the C-C, C-O, and C=O of cellulose, respectively [53]. 

SEM analysis of blank paper and ZnO@paper-4 was performed to investigate the morphology of ZnO, as illustrated in Figure 6a–d. As clearly seen from Figure 6b,d, the surface of cellulose fibers forming the paper was covered by the ZnO nanosheet. Additionally, in comparison with blank paper, the cellulose fibers of ZnO@paper-4 became rougher due to the presence of ZnO nanosheet, which may enhance the output properties of the P-TENG, as illustrated in Appendix A. Besides, the EDX elemental mapping was investigated to further confirm that the ZnO nanosheet had adhered to cellulose paper. As shown in Figure 6e–g, C, O, and Zn were uniformly distributed on the surface of cellulose fibers. The C element was assigned to cellulose composed of C, O, and H. The O element resulted from both cellulose and ZnO. The presence of Zn indicated that ZnO had successfully attached to the cellulose fibers of the paper. It also further showed the uniform distribution of ZnO on the ZnO@paper because the Zn element is homogeneously distributed.

As discussed above, the results proved that ZnO@paper could be prepared using a ZnCl_2_-urea eutectic solvent. 

### 3.2. Antibacterial Activity 

To monitor the human movement state, from hand, arm, foot, etc., the P-TENG must be in contact with human skin for a long time and exposed to the air environment, which will lead to bacterial germination. Thus, the antibacterial property of cellulose paper is an important prerequisite for a TENG used as a pressure sensor fixed on the human body to detect human motion. Furthermore, it was also reported that the bacteria in the friction material may decrease the electrical output of a TENG, impacting the effectiveness of the TENG for monitoring human motion [55,56]. Therefore, it is very important to investigate the antibacterial activity of ZnO@paper. 

ZnO is an efficient antibacterial agent that could kill Gram-positive and Gram-negative bacterial [55,57,58]. It can be seen from Table 2 that the antibacterial ratio of all ZnO@papers against *E. coli* and *S. aureus* was above 99.99%, indicating that ZnO@paper prepared in the proposed method had excellent antibacterial efficiency. As shown in Figure 7, compared to blank paper, the surviving colonies number of both *E. coli* and *S. aureus* are almost absent in the culture medium of all ZnO@papers. Therefore, it can be concluded that both *E. coli* and *S. aureus* were killed under the action of ZnO@paper. Due to the excellent antibacterial activity of ZnO@paper, the P-TENG prepared by our ZnO@paper could restrain and kill the microorganisms during the operating time. 

### 3.3. Electrical Output Performance of the P-TENG

The properties of a TENG are mainly dependent on the surface charge density of friction layers, which was affected by the roughness and triboelectricity of friction materials [59]. Herein, the ZnO content (0–20.3 wt.%) of the ZnO@paper added with the increase of ZnCl_2_ dosage in eutectic solvents, which may enhance the triboelectricity of ZnO@paper, as shown in Appendix A. In the tested range, the V_OC_ (~35 V) of the P-TENG remained unchanged when the ZnO content of the ZnO@paper was lower than 10 wt.% (the ZnCl_2_ dosage ranged from 0 to 0.03 mol); further increasing the ZnO content to 16.3 wt.% (increasing ZnCl_2_ dosage to 0.07 mol) could significantly increase the V_OC_, and the maximin V_OC_ is 77 V, as shown in Figure 8a. However, the V_OC_ decreased to 30 V when the ZnO content of the ZnO@paper was over 20.3 wt.% (the ZnCl_2_ dosage was beyond 0.1 mol). The I_SC_ increased from 0.05 μA to 0.17 μA as the ZnO content was increased from 0 to 8 wt.%, and it remained at 0.17 μA when the ZnO content ranged from 8 wt.% to 16.3 wt.% (the ZnCl_2_ dosage increases from 0.01 to 0.07 mol). However, the I_SC_ of the P-TENG declined when the ZnO content was over 20.3 wt.% (Figure 8c). The change tendency of the V_OC_ and I_SC_ may be related to the surface roughness of ZnO@paper. The high surface roughness could increase the contact area of ZnO@paper, resulting in the generation of more triboelectric charges [12,60,61]. Thus, increasing the ZnO@paper roughness may enhance the output properties of the P-TENG. The surface roughness of the ZnO@paper increased when the ZnO content was increased from 0 to 16.3 wt.%, and then decreased as the ZnO content was up to 20.3 wt.%, as shown in SEM images and Appendix A. Thus, the electrical output performance of the P-TENG decreased when the ZnO content of ZnO@paper reached 20.3 wt.%. Furthermore, to measure the effect of the external compression force applied by the linear motor on the performance of the P-TENG (prepared by ZnO@paper-4), the force was designed from 10 N to 50 N, and the V_OC_ and Isc are shown in Figure 8b,d. Increasing the external force could reduce local gaps or voids introduced by surface roughness and the shape deformation, leading to ZnO@paper and PTFE contacted more closely. Thus, the triboelectric charges would change by increasing the external force. It has been also reported that high enough triboelectric charges generate on the surface of the friction layers under the corresponding external force [62]. The results indicated clearly that the V_OC_ and I_SC_ values increased greatly when the external force was below 20 N and remained practically unchanged with further external force growth (30 N–50 N). Namely, the external force of 20 N is sufficient to produce the maximum V_OC_ and I_SC_of a P-TENG. It was mainly due to the high enough triboelectric charges on the surface of the ZnO@paper under the external force of 20 N due to the close contact of the ZnO@paper and PTFE. Further increasing the external force, the local gaps or voids introduced by surface roughness and the shape deformation could be not increased, thus the output performances of the P-TENG remain unchanged by increasing the external force to 30, 40, and 50 N.

However, the V_OC_ and I_SC_ are much lower compared to the related reports despite the ability to obtain the high output voltage and current for the P-TENG. For instance, Mi et al. fabricated a type of TENG consisting of polyamide nanofiber mats and polyimide (PI) aerogel films, which achieved a V_OC_ of 115 V and I_SC_ of 9 µA [63]. Chen et al. also demonstrated a TENG with a Voc and Isc of 196.8 V and 31.5 μA, respectively. This TENG was prepared using crepe cellulose paper and nitrocellulose membrane as the friction layers [8]. On the one hand, the releasing electron ability of paper is much weaker compared with other materials such as nylon and polyamide, leading to induce less triboelectric charges. On the other hand, the high contact-separation rate of the friction layer would produce more triboelectric charges [64]. In our study, the contact-separation frequency of the P-TENG was 1 Hz; thus, the rate may be too slow to generate less triboelectric charges, leading to the low output performance. In addition, the thickness of the friction layer is another important influence factor for the TENG. A friction layer that is too thick would increase the distance between the friction surface and the electrode; hence, a greater induction distance would yield fewer induced charges due to the edge effect of electrostatic induction [65]. The thickness of our ZnO@paper is over 125 μm, thus it may be too thick for the TENG. Fortunately, our P-TENG has the advantages of flexibility and biodegradability due to the application of the ZnO@paper made of cellulose fibers, which are flexible and biodegradable. 

### 3.4. The P-TENG-Based Pressure Sensor

To show that a P-TENG can be applied in a sensor for monitoring the human motions, the P-TENG prepared by ZnO@paper-4 was used as the typical example for the pressure sensor because its output performance is the highest. 

The P-TENG operating principle to generate electric energy in an external circuit is combining the triboelectric effect and electrostatic induction between the ZnO@paper and PTFE in periodic contact and separation. It has also been reported that the TENG sensor is based on the triboelectric effect and electrostatic induction, and it can yield capacitance, voltage, and current in response to external pressure [66,67]. Hence, the P-TENG-based pressure sensor can be in response to external pressure. It was also reported that the output signal response of the TENG sensor to pressure could be non-linearity in the whole pressure region, but it exhibited different linear with pressure in different pressure regions [68]. The sensitivity of the P-TENG-based pressure sensor was precisely carried out at 1–100 N of pressure by dropping different counterweights freely from a height of 3 cm to the central area of the pressure sensor. The peak of the response signal and output signal obtained from different pressure were given in Figure 9 and Appendix A, respectively. Figure 9 indicates that the signal response is exhibited in three distinct pressure regions: the tiny-pressure region (1–5 N) with a pressure sensitivity of 0. 0498 pF N^−1^, the low-pressure region (5–40 N) with a pressure sensitivity of 0.1811 pF N^−1^, and the high-pressure region (40–100 N) with a pressure sensitivity of 0.0488 pF N^−1^, indicating that the P-TENG-based pressure sensor possesses different sensitivity in different pressure regions.

The P-TENG-based pressure sensor has potential applications in the detection of human motion, as shown in Figure 10. Here, when a hand touched the surface of the P-TENG prepared by ZnO@paper-4, the gap between the ZnO@paper-4 and PTFE became narrow, leading to a detectable signal output, as illustrated in Figure 10a. The P-TENG was also fixed on the arm by medical tape to further demonstrate that it could detect human motion information. As the arm was repeatedly bent and relaxed, the P-TENG had contact and separation, generating the responded signal, as presented in Figure 10b. As shown in the inset of Figure 10c, the P-TENG-based sensor was located on the back of a t-shirt and further employed to harvest the mechanical energy while a human was leaning back on a wood chair. Consequently, the P-TENG-based sensor exhibited a good response signal. This phenomenon also occurs in the process of walking, which is shown in Figure 10d. The fabricated P-TENG-based sensor was placed on the sole of a foot to monitor its movement. When the foot stands alone, the response signal generated by human body pressure reaches the highest value. If the foot is raised, the sensor signal value returns to the microampere level. The results indicated that not only can our P-TENG harvest energy from human motion, but also it can also be applied as a sensor for obtaining the human motion information.

## 4. Conclusions

In the present work, a facile method has been successfully demonstrated to fabricate ZnO@paper. In the proposed method, ZnCl_2_-urea eutectic was first introduced to swell and cleave the hydrogen bonds in the cellulose structure by water destruction; while the exposed hydroxyl group would be hydroxyl bonded with ZnO that was converted by the dehydration of Zn(OH)_4_^2−^, which endowed the ZnO@paper with excellent antibacterial activity. In addition, the ZnO@paper also has potential application in the P-TENG as a friction layer and in print paper as a substrate, which generates the maximum V_OC_ of 77 V and I_SC_ of 0.17 μA by harvesting external force. The results indicate that the presence of nanosheet ZnO in cellulose paper is beneficial for the electrical output performance of a P-TENG. Furthermore, the P-TENG prepared using ZnO@paper could impede the accumulation of bacteria during the longtime work process owing to the excellent antibacterial activity of ZnO@paper, which could be applied in a wearable sensor for harvesting energy from human motion and detecting human motion information.

## Figures and Tables

**Figure 1 nanomaterials-11-01111-f001:**
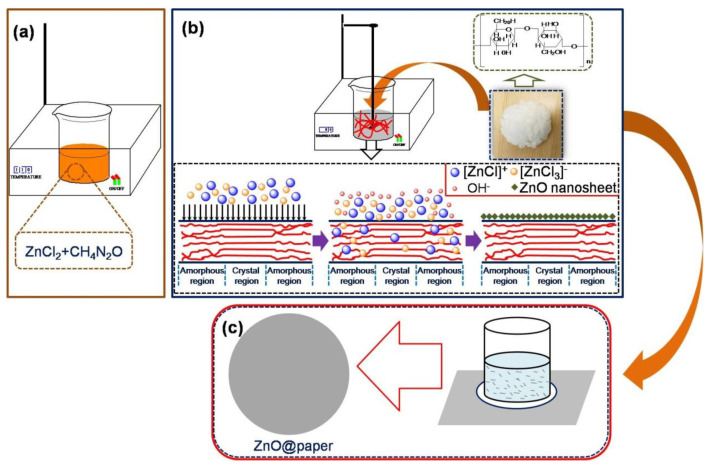
The preparation process of ZnO@paper. (**a**) The fabrication of eutectic solvent; (**b**) The ZnO-modified cellulose fibers; (**c**) The ZnO@paper preparation.

**Figure 2 nanomaterials-11-01111-f002:**
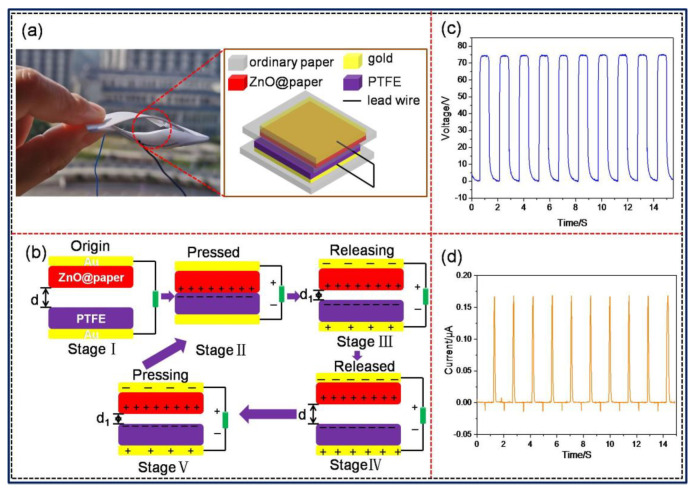
The structure, working mechanism, and electrical outputs of the P-TENG in a vertical contact-separation mode. (**a**) The structure and (**b**) schematic working principle of the P-TENG. Electrical outputs of the P-TENG: (**c**) open-circuit voltage (V_OC_) and (**d**) short-circuit current (I_SC_).

**Figure 3 nanomaterials-11-01111-f003:**
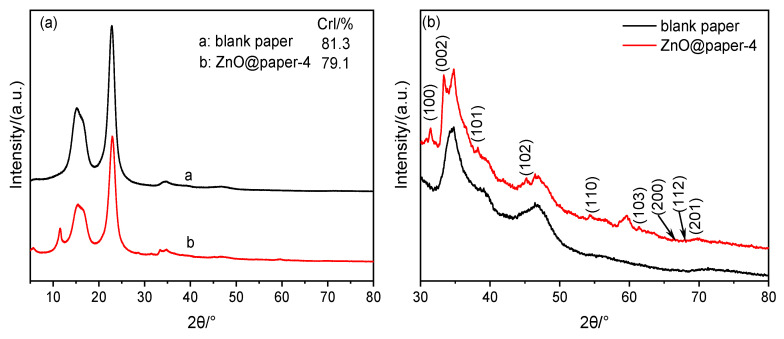
XRD patterns of blank paper and ZnO@paper. (**a**) total spectrogram; (**b**) partial spectrogram.

**Figure 4 nanomaterials-11-01111-f004:**
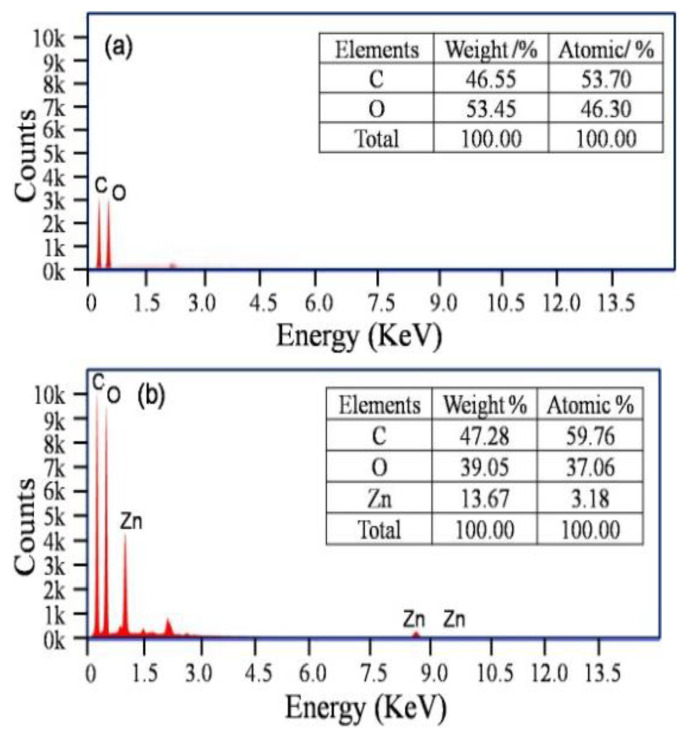
EDS patterns. (**a**) blank paper; (**b**) ZnO@paper-4.

**Figure 5 nanomaterials-11-01111-f005:**
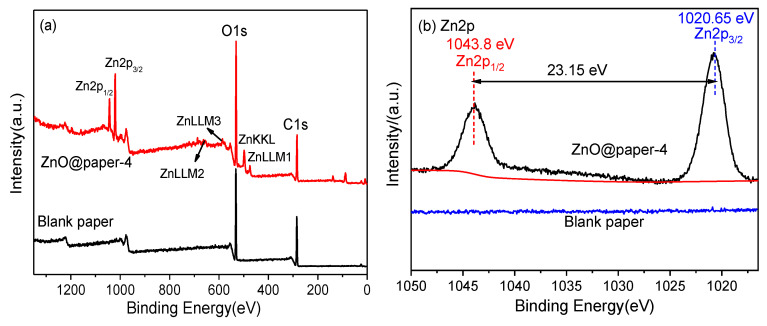
XPS spectra of blank paper and ZnO@paper-4: (**a**) survey scan; (**b**) high-resolution Zn_2_p spectra; (**c**) O1s peaks, (**d**) C1s peak of blank paper and ZnO@paper-4.

**Figure 6 nanomaterials-11-01111-f006:**
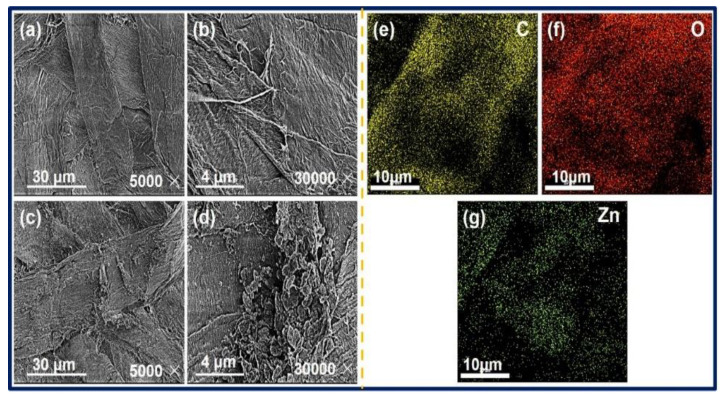
SEM image of blank paper (**a**,**b**), ZnO@paper-4 (**c**,**d**); EDX elemental mapping results of ZnO@paper-4 (**e**–**g**)**.**

**Figure 7 nanomaterials-11-01111-f007:**
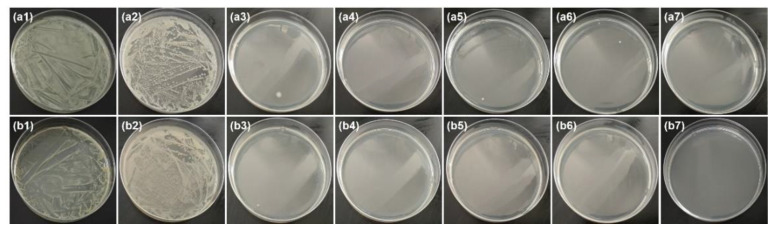
Culture medium under bacterial suspensions diluted 100 times. From left to right, the sample is the control group, blank paper, ZnO@paper-1, ZnO@paper-2, ZnO@paper-3, ZnO@paper-4, and ZnO@paper-5, respectively. (**a**) *E. coli*, (**b**) *S. aureus*.

**Figure 8 nanomaterials-11-01111-f008:**
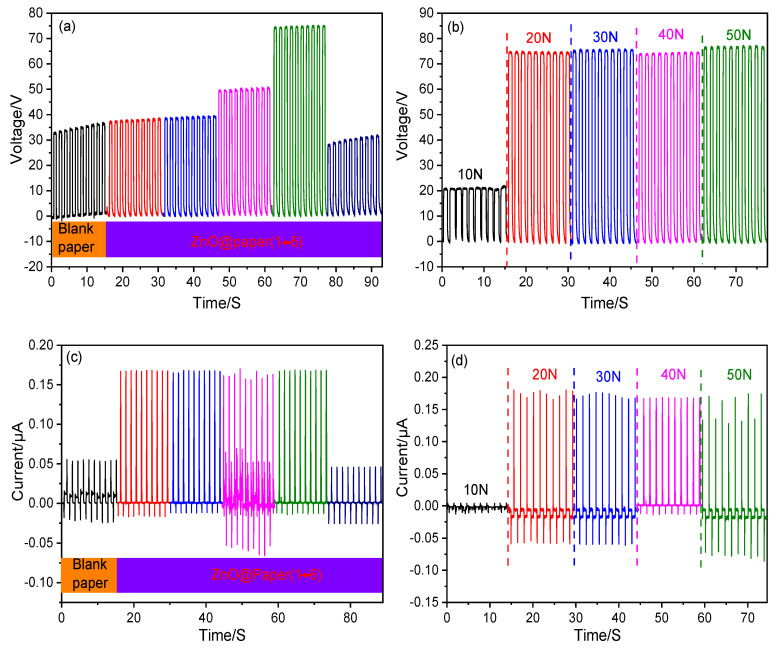
The electrical output performance of the P-TENG fabricated by ZnO@paper. (**a**,**c**) V_OC_ and I_SC_ of the P-TENG; (**b**,**d**) V_OC_ and I_SC_ at variable external force.

**Figure 9 nanomaterials-11-01111-f009:**
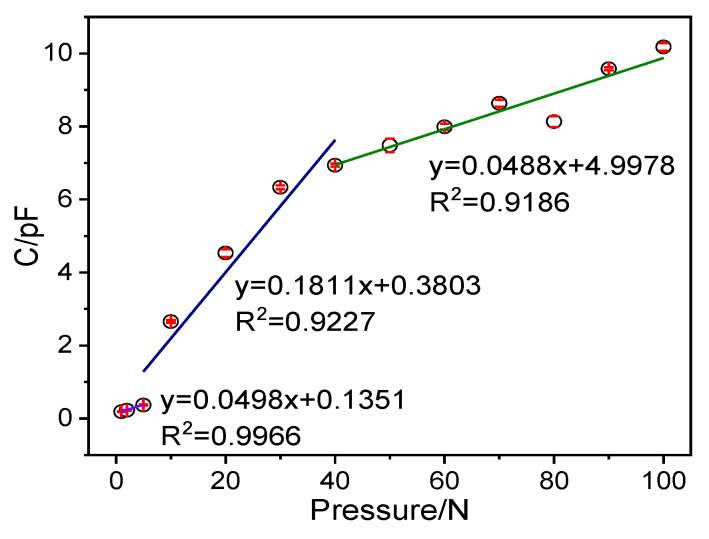
The maximum peak of the output signal of the pressure sensor in response to applied pressure.

**Figure 10 nanomaterials-11-01111-f010:**
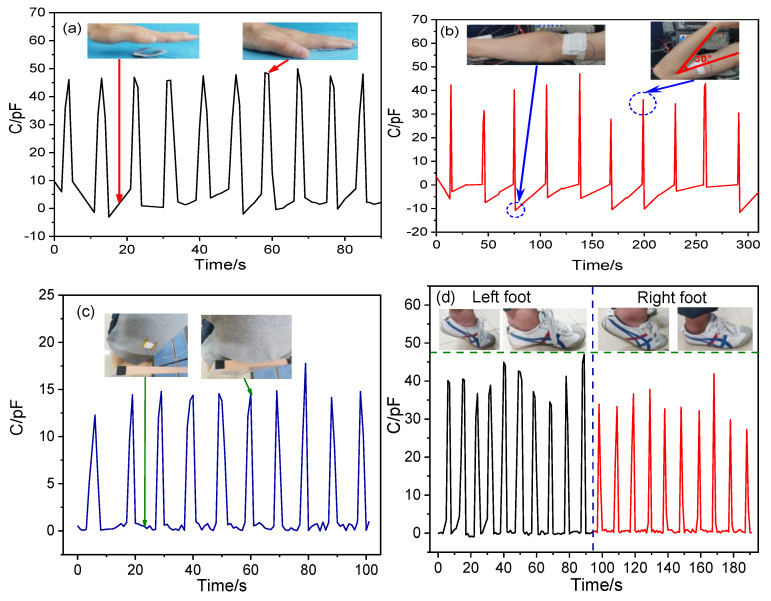
Application of the P-TENG-based pressure sensor in human motion information such as (**a**) hand flapping, (**b**) leaning back, (**c**) arm bending, and (**d**) walking.

**Table 1 nanomaterials-11-01111-t001:** Summary of studies investigating various preparation methods of ZnO/cellulose composite.

Materials	Preparation Method	C/(wt.%)	O/(wt.%)	Zn/(wt.%)	Ref.
Filter paper, ZnCl_2_, NaOH	Precipitation	60.17	37.47	2.1	[36]
Cotton pulp, ZnO nanoparticles	Blending	42.57	48.97	6.72	[32]
CNC, ZnCl_2_, NaOH	Precipitation	57.97	35	7.02	[48]
Dissolving pulp, ZnCl_2_, CO(NH_2_)_2_		47.28	39.05	13.67	Our work

**Table 2 nanomaterials-11-01111-t002:** The CFU of the control group, blank paper, and ZnO@paper, and antibacterial ratio of ZnO@paper tested against *E. coli* and *S. aureus.*

	*E. coli* (Gram^−^)		*S. aureus* (Gram^+^)	
	CFU	Antibacterial Ratio (%)	CFU	Antibacterial Ratio (%)
Control group	103 × 10^7^	0	109 × 10^6^	0
Blank paper	160 × 10^7^	0	102 × 10^6^	0
ZnO@paper-1	1 × 10^2^	100	1 × 10^2^	99.99
ZnO@paper-2	0	100	0	100
ZnO@paper-3	0	100	0	100
ZnO@paper-4	0	100	0	100
ZnO@paper-5	0	100	0	100

## Data Availability

The data presented in this study are available on request from the corresponding author.

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
