# Peer review of "Cellulose Paper Modified by a Zinc Oxide Nanosheet Using a ZnCl2-Urea Eutectic Solvent for Novel Applications"

_nanomaterials, 2021, doi:10.3390/nano11051111_

Round 1

Reviewer 1 Report

In the submitted manuscript the authors considered the use of ZnO nanosheet as cellulose modifier for the preparation of triboelectronic generators.  The resulting properties sound nice and enlarge the possible final applications of the prototype, so the idea is convincing and deserves to be considered for publication in the journal. In order to improve the already good quality of the manuscript, the following items should be considered:

  • English language needs to be improved (see, for example, defibration devise, testify, cellulose swelling cellulose..);
  • in table 2,results of antibacterial tests have been reported for ZnO@paper-1, while all the other characterization have been provided for ZnO@paper-4; please, insert the results for this material coherently.

Reviewer 2 Report

The authors presented an investigation of the production and characterisation of zinc oxide nanosheet modified paper and its applications in pressure sensors and as antibacterial material.

The work presented by the authors is interesting and has some value. Before the manuscript can be accepted for publication, the following points should be addressed. Additionally, the manuscript should be checked for typos; the quality of English is quite poor and must be improved.

Abstract

Line 12: When the authors say that “cellulose paper has been functionalised” do they mean by other groups? What type of functionalisation are they referring to? Clarify

Line 15: The authors use the word “facile” which is English may have a negative and diminishing meaning. They should use a more suitable word instead.

Line 16: Are the “cellulose fibres” the nanosheets? Clarify

Lines 18 – 19: Was the production of the flexible P-TENG the main goal of the research? Clarify

Line 23: When the authors talk about “modification of ZnO” do they mean modification of the nanosheets which are made of the cellulose nanofibers? Clarify

Line 24: The sentence “suggesting that the P-TENG features with antibacterial activity” is incomplete and need to be amended.

Introduction

Line 46: What is “they”? Clarify

Line 54: Explain what type “flexibility” is meant in this context

Line 55: The sentence “composite paper …. cellulose fibres” seems to be incomplete and needs to be amended

Line 56: What is “composed of paper”? Clarify

Line 56: What “is neglected”? Clarify

Line 65: It is unclear what is meant by “paper or cellulose fibres”. Are the authors referring to either paper, which is made of cellulose fibres, or individual cellulose fibres? Clarify

Line 67: Do the authors mean the paper after immersion in the Zn solution? Clarify

Line 69: The sentence “so that the nucleation …. on cellulose fibres” is incomplete because the verb is missing. Amend it

Line 72: What does “poor accessibility” means in this context?

Line 73: What do the authors mean by “limiting the ZnO … industrial production”? Is not scaling up one of the goals? Clarify

Line 76: Give examples of the “versatile application”

Line 77: Is the use of the ZnO@paper as “friction layer for P-TENG” the main goal of this research? Clarify

Line 79: The authors mention “pressure sensor” as an application of the ZnO@paper; what about the “antibacterial activity” and its applications?

Lines 80 – 81: Should not the “formation mechanism …. on cellulose” be the first goal of this research, followed by applications?  

Materials

Line 84: Clarify what “dissolving pulp” is meant in this context

Line 89: Heading “Integration of the P-TENG” is unclear. It is not clear what type integration and into what.

Line 90: How thick were the ZnO@paper and the PTFE? Quantify

Line 91: “Was” should be changed into “were”

Line 91: How thin was the gold film? Quantify

Line 92: What were the parameters used for sputtering? Give details

Line 93: What does “gold film of ZnO@paper” mean? Was this the gold film sputtered onto the ZnO@paper?

Line 93: What kind of “print paper”? Clarify

Line 94: How was the “PTFE film pasted”? Did they use glue? Give details

Line 95: What are the “two pieces” mentioned? Are they the ZnO@paper and the PTFE? Clarify

Line 95: What does “fixed” mean? Does it mean “joint together”? Clarify

Line 95: Why was “Kapton tape” used? Why not 3M tape as previously done? Clarify

Line 96: What does “beneficial for contact-separation of P-TENG” mean? What was beneficial? Was the shape beneficial, why? Clarify

Line 97: When the authors refer to “pressure sensor” do they mean to say that the pressure sensor was a P-TENG but just smaller? Clarify

Line 108: What does “fully contacted” mean?

Line 120: Should the section 2.3 come before the section 2.2?

Line 121: What type of “dissolving pulp”?

Line 126: Was the stirring of the suspension done at room temperature or at 55C? Clarify

Line 127: Why was the temperature increased to 85C? Explain

Line 128: How many times was the reaction product washed with distilled water?

Line 128: How was the size of 60g/m2 controlled?

Line 129: How was the ZnO@paper dried “by squeeze”?

Line 131: Is the “cellulose paper” the “blank” paper?

Line 131: How were the different amounts of ZnCl2 chosen?

Line 132: Was the amount of CO(NH2)2 also adjusted so that the 1:3 ratio was maintained?

Line 135: What was XRD used for?

Line 138: What were the parameters used for SEM?

Line 140: The “chemical bonding states” were between what?

Line 144: “applied” should be changed into “used”

Line 145: Give details of the preparation of culture medium

Line 146: What does LB stand for?

Line 146: What kind of “apparatus” was sterilised?

Line 146: How was the sterilisation carried out? Give details/parameter

Line 148: Give quantity/concentration of E. Coli and S. aureus added into the LB broth

Lines 148 – 149: What does “containing sample … small pieces” mean?

Line 151: What “different dilutions” are these?

Line 152: What does “to account directly” mean?

Line 153: Why were the suspensions “diluted 100 times”? What was the concentration used? Explain

Lines 153 – 154: What does “fully distributed among the culture medium” mean?

Line 154: How was the counting of the CFUs carried out? Explain

Line 155: What does “from the appropriate dilution” mean? Quantify the dilution used

Line 157: How was the “blank paper” prepared?

Line 157: Equation should be numbered

Line 158: The heading of section 2.6 should have “as” instead of “and”

Line 160: Quantify “very large”

Lines 162 – 163: What does “to provide at … the P-TENG” mean?

Line 164: What are the “relevant data” mentioned? Explain

Line 165: How was the “pressure sensor assembled by P-TENG”? Give details

What about data analysis? How were the data analysed? What type of statistical analysis was carried out? Give details   

Results and discussion

Line 168: Change word “facile” (see previous comment)

The content of section 3.1 should be moved to the section “Methods” because it described the method used to prepare ZnO@paper

Line 201: Only ZnO@paper-4 is mentioned. What about the other ZnO@papers? Were they used? What are results?

Line 202: Show the spectrum of cellulose I for comparison

Lines 204 – 205: Why is it important that “the crystalline index of ZnO@paper-4 is lower than blank paper”? Explain

Line 211: Does the “hexagonal structure” of ZNO paper have any impact on the results obtained?

Line 255: What were the dimensions of the ZnO sheets observed in Fig. 6? Were those sheets monolayers or multilayers? Were they homogenously distributed?  

Line 256: Quantify “rougher”. How was the roughness measured? Was the roughness measure using other techniques, e.g. Atomic Force Microscopy?

Line 257: How does increase in roughness “enhance the output properties”?

Line 259: How did EDX mapping confirmed the adherence of ZnO nanosheets to paper? Explain and give details

Line 260: Fig. 6 does not seem to show that C, O, and Zn were uniformly distributed, in particular because the images are so dark that it is very difficult to discriminate the picture from the background. Contrast must be improved

Lines 260 – 261: From where the C and O elements were obtained? Explain

Line 271: What kind of “human movements” and what type of “information”?

Line 274: “Used as pressure” should be changed into “used as sensor” or “used as pressure sensor”

Line 274: How was “human motion” detected using this pressure sensor? Give details

Line 280: Quantify “certain dosage”

Table 2: What was the “control group” used? Why was ZnO@paper-1 mentioned but not the others? Do the straight lines in the column for the antibacterial ratio mean 0%?

Line 299: Quantify “changed slightly”

Lines 300 -  305: It is unclear why that information is relevant

Line 307: Why is the triboelectricity of ZnO@paper enhanced? Was this tested using other techniques, e.g. electrostatic force microscopy?

Line 310: How does the decrease in electrical output link to surface roughness? Were different techniques used to verify and confirm that link?

Line 312: How was “external compression force” applied?

Lines 318 – 320: What does “it is better … layers of P-TENG” mean?

Lines 320 – 321: Why do triboelectric charges changes by increasing the external force? How “slightly” do they increase?

Lines 322 – 323: What are these discrepancies due to?

Lines 332 – 333: What that tested using other frequencies? Why was 1Hz chosen?

Line 335: How “thick”? Quantify

Line 338: Was the thickness of the ZnO@paper decreased to verify such an assumption?

Line 339: Why does ZnO@paper makes P-TENG flexible and biodegradable? Explain

Lines 340 – 343: This sentence should be moved to the section “Conclusions”

Line 358: How was the sensitivity measured?

Figure 9: How was the graph obtained? Experimental points do not have error bars. How were the experimental data interpolated? What type do data analysis was carried out?

Line 370: Only ZnO@paper-4 is mentioned. What about the other types of ZnO papers?    

Reviewer 3 Report

This is a very nice publication carefully put together, well written and well explained, with lots of work and experiments having been done, and lots of useful data. A very complete manuscript. Only part 3.2 is a little bit confusing since it is not clear why two different methods of characterizations of ZnO@paper are reported. and how this EDS is related to the XPS. Also, figure 8 seems to be confusing since it is related only to the leading to the low output performance. Please clarify.

Round 2

Reviewer 1 Report

The authors considered the sugegsted modifications, so now the paper can be considered for publication in its present form

Reviewer 2 Report

Thank you for your replies to the comments and questions. All of the comments in the reviewer's report were meant to help the authors revise the manuscript. Although the authors amended some parts of the manuscript, other parts have been left as they were. The authors should incorporate all the answers given to the reviewer's report in the manuscript.

Reviewer 3 Report

The article can be accepted for publication.
